# Effect of Ethanol on Preparation of Konjac Emulgel-Based Fat Analogue by Freeze-Thaw Treatment

**DOI:** 10.3390/foods11203173

**Published:** 2022-10-11

**Authors:** Jie Jiang, Abel Wend-Soo Zongo, Fang Geng, Jing Li, Bin Li

**Affiliations:** 1College of Food Science and Technology, Huazhong Agricultural University, Wuhan 430070, China; 2Key Laboratory of Environment Correlative Dietology (Huazhong Agricultural University), Ministry of Education, Wuhan 430070, China; 3College of Food and Biological Engineering, Chengdu University, No. 2025 Chengluo Avenue, Chengdu 610106, China

**Keywords:** konjac emulgel, ethanol, pork backfat, fat analogue, mechanical textural properties, physicochemical properties

## Abstract

In the current study, a method using ethanol to modulate the texture properties of konjac gel during freeze-thaw process was used to prepare konjac emulgel-based fat analogue. A certain amount of ethanol was added to konjac emulsion, heated to form a konjac emulgel, then frozen at −18 °C for 24 h, and finally thawed to obtain konjac emulgel-based fat analogue. The effects of different ethanol contents on the properties of frozen konjac emulgel were explored, and data was analyzed by one-way analysis of variance (ANOVA). The emulgels were compared with pork backfat in terms of hardness, chewiness, tenderness, gel strength, pH, and color. The results showed that the konjac emulgel with 6% ethanol had similar mechanical and physicochemical properties to pork backfat after freeze-thaw treatment. The results of syneresis rate and SEM showed that adding 6% ethanol could not only reduce the syneresis rate, but also effectively weaken the damage to the network structure caused by freeze-thaw treatment. The pH value of konjac emulgel-based fat analogue was between 8.35–8.76, and the L* value was similar to that of pork backfat. The addition of ethanol provided a new idea for the preparation of fat analogues.

## 1. Introduction

As one of the main component in foods, fat plays an essential part in foods’ texture and sensory characteristics [1]. However, high-fat foods with low fiber have brought many health problems [2]. Consumers’ demand for healthy foods has led to a trend of increasing demand for low-fat products [3]. Therefore, the health industry is seeking alternatives to substitute some of the fat in some foods without reducing the taste [4].

Konjac glucomannan (KGM) is a neutral water-soluble polysaccharide extracted from the tuber of *Amorphophallus konjac* [5], composed of β-1,4 linked D-mannose and D-glucose at a molar ratio of 1.6:1 or 1.4:1. KGM has many good functional properties, such as gelation, thickening [6], and can form thermally irreversible gels under certain conditions [7]. KGM has been reported to be good for human health and can also cure some diseases [8]. The physicochemical properties of KGM make it potentially useful as a fat analogue.

Many studies have examined the effectiveness of KGM as a fat analogue [9,10,11,12], and have mostly focused on its use as a fat replacement in minced meat or sausages. However, few studies have investigated konjac gel as a fat analogue compared to pork backfat. Some studies have reported that the combination of konjac gel and healthy oils (fish oil, flaxseed oil, olive oil) used as a fat simulant to replace the fat in frankfurters can reduce the fat content [13]. Jiménez-Colmenero et al. used konjac flour, carrageenan, and starch to prepare a fat analogue to simulate the texture characteristics of pork backfat, but its texture is quite different from pork backfat [14]. Konjac gel was an ideal material for simulating pork backfat, but its texture was soft and had a certain gap with pork backfat. The freeze-thaw treatment was an effective method to improve the texture of konjac gel, but it seriously damages the network structure of the gel, resulting in severe water separation and abnormal increase in gel hardness [15].

In this study, we tried to combine a poor solvent (ethanol) with a good solvent (water) to affect the assembly during the freezing process by influencing the state of the glucomannan molecular chain. The treatment used in this study was that a certain amount of ethanol was added to konjac sol, heated to form an emulgel, then frozen at −18 °C for 24 h, and finally thawed to obtain a konjac emulgel-based fat analogue. The freeze-thaw treatment for konjac emulgel was only one cycle in this study, because multiple cycles would lead to severe water separation and damage to the network structure. The purpose of this research was to prepare a konjac emulgel that was close to pork backfat and to study its physical, chemical, and mechanical properties. This study provided a simple and efficient method to simulate pork backfat, and also provided a new case for the combination of poor and good solvents to affect polysaccharide gels.

## 2. Materials and Methods

### 2.1. Materials

Konjac glucomannan (Mw = 650 kDa) was purchased from Hubei Konson Konjac Technology Co., Ltd. (Wuhan, China). Soybean oil produced by Yihai Kerry Foods (Wuhan, China) was purchased from a local market. Ethanol was purchased from Henan Xinheyang Alcohol Co., Ltd. (Mengzhou, China). All other chemicals used were of analytical grade. 

### 2.2. Preparation of Konjac Emulgels

Na_2_CO_3_ (0.16% *w*/*w*) solutions were prepared by stirring continuously 1 h at room temperature. The ethanol/soybean oil mixed solutions were prepared by mixing an appropriate amount of ethanol and soybean oil under magnetic stirring (B13-3 and Shanghai Sile Instrument Co., Ltd., Shanghai, China). KGM was added into the Na_2_CO_3_ solution, and then the ethanol/soybean oil mixed solution was immediately added with mechanical stirring (300 rpm) to prepare a emulsion. The KGM concentration in the emulsion was 4.0 wt%. The ethanol concentrations in the emulsion were 0.0 wt%, 2.0 wt%, 6.0 wt% and 8.0 wt%. The emulsion was poured into a mold with dimensions of 170 mm × 95 mm × 20 mm and stored at room temperature for 4 h, then heated at 90 °C for 1 h in a water bath to form konjac emulgel. When the core temperature of the emulgel reached room temperature, emulgels were frozen at −18 °C for 24 h, and finally taken out and thawed at room temperature for 4 h.

### 2.3. Water and Fat Binding Properties

The initial weight of the sample was weighed and recorded as m_0_, and then the sample was placed in a tube and heated at 70 °C for 30 min in a water bath. After heating, take out the emulgel and pour out the water and fat, then weigh the emulgel again and record as m. Heating loss (water and fat loss) was calculated by the following equation:$$\mathrm{Heating}\;\mathrm{loss}\left(\%\right)=(m-m_{0})/m_{0}\;\times \;100$$

### 2.4. Syneresis Rate Measurement

The weight of the samples before freezing was recorded as A (g), and the weight of the samples after freezing was recorded as B (g). The water separation rate can be calculated by the following formula. The syneresis rate was calculated by the following equation:Syneresis (%) = (A − B)/A × 100

### 2.5. Measurement of pH

The samples were mixed with distilled water at a ratio of 1:10 (g/mL), then filtered with a funnel. Finally, the pH value was measured at room temperature with a Radiometer model PHM 93 pH meter (Meterlab, Copenhagen, Denmark).

### 2.6. Colour Measurement

Colour parameters, such as L* value, a* value and b* value of the samples, were measured on a chromameter (UItraScan VIS HunterLab, USA). According to the following equations, the whiteness index and browning index were calculated, which represents how pure white was and brown was [16].
$$\Delta E=\sqrt{{\left({b-b}_{\mathrm{pork}\;\mathrm{backfat}}\right)}^{2}+{\left({a-a}_{\mathrm{pork}\;\mathrm{backfat}}\right)}^{2}+{\left({L-L}_{\mathrm{pork}\;\mathrm{backfat}}\right)}^{2}}\;\mathrm{Whiteness}\;\mathrm{Index}=100-\sqrt{{\left(100-L\right)}^{2}{+b}^{2}{+a}^{2}}\;\mathrm{chroma}=\sqrt{\left(a^{2}{+b}^{2}\right)}\;\mathrm{Browning}\;\mathrm{Index}=\frac{\left[\left(X-0.31\right)\times 100\right]}{0.172}\;X=\frac{a+1.75L}{5.645L+a-3.012b}$$

### 2.7. Mechanical Properties

Texture Profile Analysis (TPA), puncture test, and shear test were performed on the samples using the same texture analyzer (TA-XT plus, Stable Micro Systems Ltd., Godalming, UK).

Texture attributes were obtained by TPA, and the aluminum cylindrical probe (SMP P/36 R) was applied during the test. The sample dimension is 30 mm × 30 mm × 20 mm. Samples were axially compressed to 40% strain at a pre-test speed of 1 mm/s and a post-test speed of 5 mm/s [17].

The corresponding force-penetration curves were obtained at a crosshead speed of 1 mm/s and analyzed in puncture test. The load (as breaking force) when the emulgel sample lost its strength and ruptured was recorded.

The shear test was carried out using a HDP-BSW device, which is a blade. The force was exerted to a compression distance of 10 mm at 1 mm/s crosshead speed. The shear value was calculated as the maximum shear force of the blade.

### 2.8. Low Field NMR Transverse Relaxation (T_2_) Measurements (LF-NMR)

The LF-NMR relaxation measurements were carried out using a low field NMR analyzer (MesoQMR23-060H, Niumag Electric Corporation, Shanghai, China). The transverse relaxation time (T_2_) was measured using the Carr–Purcell–Meiboom–Gill (CPMG) sequence. 

### 2.9. Morphology Observation

The samples were put into liquid nitrogen for fast freezing and then placed in a vacuum freeze dryer to obtain lyophilized samples. The samples were sprayed with gold using an ion sputtering device under low vacuum conditions. Finally, the surface morphology of the samples was observed by scanning electron microscopy (JSM-6390LV, Jeol, Akishima, Japan) at an accelerating voltage of 5 Kv and a magnification of 100×.

### 2.10. Statistical Analysis

The results were given as mean ± standard deviation (SD) and using SPSS 22.0 for statistical analyses. The data were analyzed using one-way analysis of variance (ANOVA), and Duncan’s test was used to determine the difference between the means at the significance level of 0.05.

## 3. Results and Discussion

### 3.1. Syneresis Rate, Heating Loss and pH

Hydrogels undergo syneresis after freeze-thaw treatment, which is a common phenomenon [18]. It can be seen from Table 1 that when the ethanol content was 6%, the syneresis rate was the lowest. As an antifreeze agent, ethanol can improve the antifreeze performance of the gel, reduce the damage to the gel network structure caused by freeze-thaw treatment, thereby reducing the syneresis rate of the gel. However, ethanol was a poor solvent for konjac gel, and high ethanol content was not conducive to the formation of the konjac emulgel network structure. Hence, the syneresis rate increased when the ethanol content was 8%.

Pork backfat and fat analogues are mostly used in the production of cooked meat products. Therefore, the water and fat binding characteristics of the products during the heat treatment process are of great significance to the processing of meat products [14]. As shown in Table 1, when the ethanol content was 0%, the heating loss of the emulgel was the largest, which may be because the freeze-thaw process severely damaged the network structure of the emulgel, so water and fat were easily lost during the heating process. The addition of ethanol can significantly reduce the heating loss of the emulgel and improve the thermal processing stability of the emulgel.

The pH value of pork backfat was 6.30 ± 0.08, and konjac emulgel was 8.35 ± 0.04 to 8.76 ± 0.06. The pH of konjac emulgel was higher than that of pork backfat.

### 3.2. Colour Analysis

Food choice is significantly influenced by colour because it influences consumers’ preference and acceptance [19]. As can be seen from Table 2, the L* value of the frozen emulgel with an ethanol content of 6% was close to that of pork backfat. However, the a* and b* values of all emulgels were quite different from pork backfat. The change in L* value of emulgel caused by the freeze-thaw treatment may be related to water loss.

### 3.3. Mechanical Properties

The mechanical properties of foods influence how consumers perceive their textural characteristics [20]. It is well known that the freeze-thaw treatment is an effective method to modify the mechanical properties of gels [21]. The crystallites formed during freezing strengthened and instigated more interactions among the biopolymer molecules, resulting in a denser and more aggregated polymer network, which, in turn, strengthens the structure [22]. As shown in Figure 1, the hardness and chewiness of the emulgel were greatly increased after freezing, but that of the emulgel with 6% ethanol increased less, indicating that the amount of ethanol would improve the freeze-thaw stability of the konjac emulgel. The texture of frozen konjac emulgel with 6% ethanol was similar to pork backfat. The springiness change of the emulgel after freezing was small, and the cohesiveness was increased to a certain extent, which may be related to the association of polysaccharide molecules.

It can be seen from Figure 2 that all unfrozen emulgels had a shear value lower than pork backfat. All samples showed increased shear values after freezing. The shear force value of the frozen emulgel was significantly greater than that of pork backfat, except for the emulgel with 8% ethanol content.

As shown in Figure 3, the breaking force of the unfrozen emulgel was significantly lower than that of pork backfat; however, the breaking force increased after freezing. Compared with other samples, the increase in breaking force of the sample with 6% ethanol was the lowest. The breaking force of the emulgel with an ethanol content of 6% after freezing was not significantly different from that of pork backfat. 

### 3.4. LF-NMR Analysis

LF-NMR is a testing technique that can be used to measurement the degree of freedom and distribution of water in gel systems [23]. The shorter the relaxation time, the tighter the binding of water molecules and the weaker the mobility of water molecules [24]. T_2b_, T_21_, T_22_, and T_23_ were the relaxation times of different states of water in the gel. T_2b_ represents water closely related to macromolecules with a relaxation time of 1–10 ms, T_21_ represents water that does not flow easily, with a relaxation time of 10–100 ms, T_22_ and T_23_ represent free water, with a relaxation time of 100–1000 ms and 1000–10,000 ms.

As shown in Figure 4, a stronger signal was observed for the T_22_ component when the ethanol content was 0 wt%, which indicated that the free water ratio of the konjac emulgel-based fat analogue was higher at this time. However, the signal intensity of the T_22_ component was significantly reduced after adding 2 wt% ethanol, indicating that the different state water has changed. In addition, it can be seen that the signal intensity of T_22_ component decreases continuously, and T_21_ and T_22_ move to a lower relaxation time with the increase of ethanol content, indicating that the fluidity of water in the konjac emulgel-based fat analogue had decreased and free water gradually turned into bound water.

The proportion of the T_2_ distribution of konjac emulgel are shown in Figure 5. P_21_ in the unfrozen emulgels increased gradually with the increase of ethanol content, while P_22_ decreased, indicating that ethanol can change the water distribution, and free water was continuously converted into immobilized water. When the ethanol contents were 0 wt% and 2 wt%, P_23_ increased significantly, and P_21_ decreased after freezing, indicating that the degree of freedom of water increased and water migrated to free water. However, when the ethanol contents were 6 wt% and 8 wt%, P_23_ did not increase significantly after freezing, and the change in water distribution was small, indicating that konjac emulgel-based fat analogue prepared with this ethanol content had better stability.

### 3.5. Morphological Analysis

As observed from the Figure 6, the pores of the emulgels with ethanol content of 0 wt% and 2 wt% became significantly larger after freezing, indicating that freezing will seriously damage the network structure of the emulgels. However, when the ethanol content was 6 wt%, the emulgel network changed less after freeze-thaw treatment, indicating that a certain content of ethanol can reduce the damage to the network structure caused by freeze-thaw treatment. The current result was in line with other measurement results in this study.

## 4. Conclusions

The current study results showed that it is feasible to use ethanol to modulate the textural properties of konjac emulgel during freeze-thaw processes for the preparation of konjac emulgel-based fat analogue. The method solved the problem of the gap between the texture of traditional konjac emulgel and pork backfat. Adding 6% ethanol and combining with one freeze-thaw treatment cycle can obtain an emulgel with similar texture properties to pork backfat, and its mechanical textural properties such as hardness, chewiness, and gel strength were similar to those of pork backfat. At the same time, the konjac emulgel-based fat analogue also had good thermal stability with a small heating loss of 4.56%. Whether this scheme has the properties of generalization remain to be revealed, and the specific mechanism of changes in emulgelation and freezing remains to be revealed in the next step.

## Figures and Tables

**Figure 1 foods-11-03173-f001:**
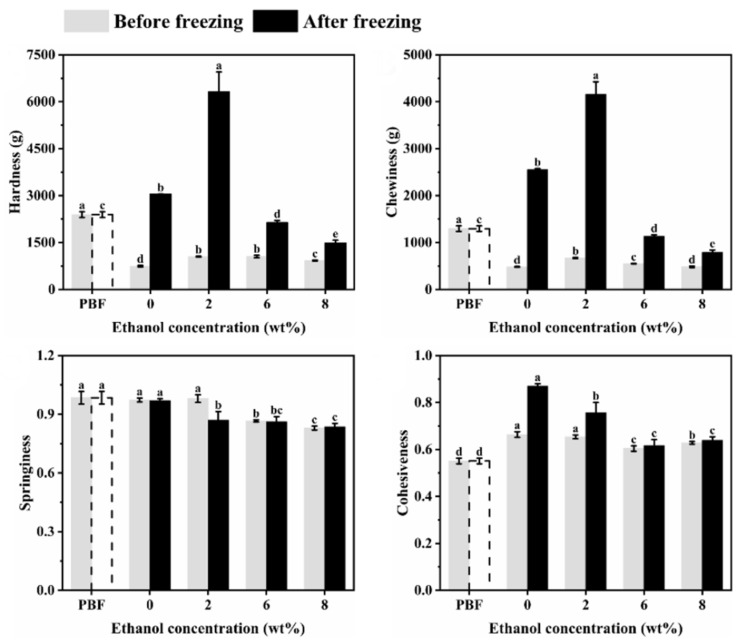
Textural profiles analysis (TPA) of pork backfat (PBF) and konjac emulgel-based fat analogue (KGFA) with different ethanol content before and after freeze-thaw treatment. Dotted line indicates that PBF was not frozen. Different letters indicate significant differences in columns of the same color (*p* < 0.05).

**Figure 2 foods-11-03173-f002:**
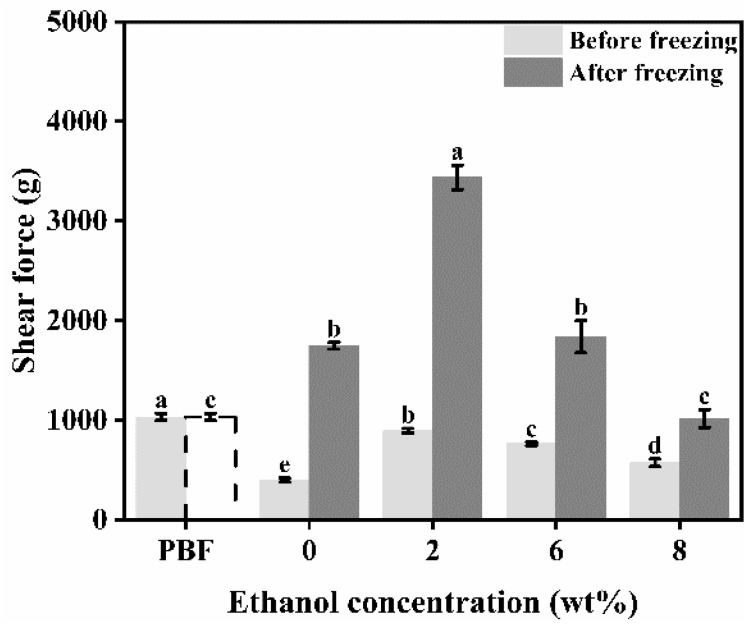
Shear force of pork back fat (PBF) and konjac emulgel-based fat analogue (KGFA) with different ethanol content before and after freeze-thaw treatment. Dotted line indicates that PBF was not frozen. Different letters indicate significant differences in columns of the same color (*p* < 0.05).

**Figure 3 foods-11-03173-f003:**
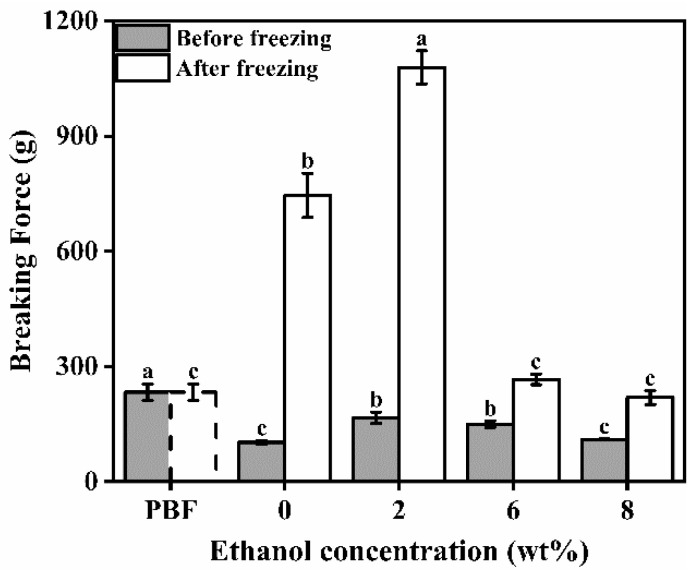
Breaking force of pork back fat (PBF) and konjac emulgel-based fat analogue (KGFA) with different ethanol content before and after freeze-thaw treatment. Dotted line indicates that PBF was not frozen. Different letters indicate significant differences in columns of the same color (*p* < 0.05).

**Figure 4 foods-11-03173-f004:**
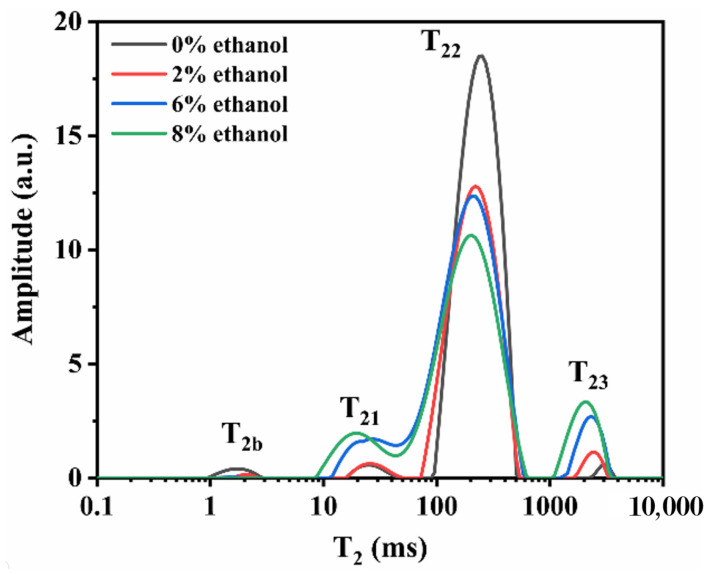
Distributions of T_2_ relaxation times of konjac emulgel-based fat analogue (KGFA) with different ethanol content without freeze-thaw treatment.

**Figure 5 foods-11-03173-f005:**
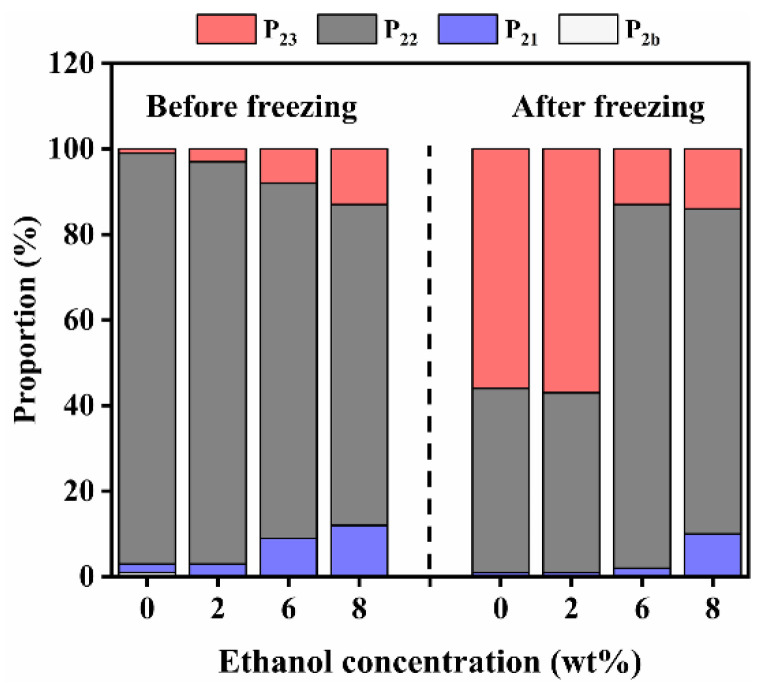
The proportion of water distribution of konjac emulgel-based fat analogue (KGFA) with different ethanol before and after freeze-thaw treatment. P_2b_, P_21_, P_22_ and P_23_ were the corresponding area fractions of T_2b_, T_21_, T_22_ and T_23_ in Figure 4.

**Figure 6 foods-11-03173-f006:**
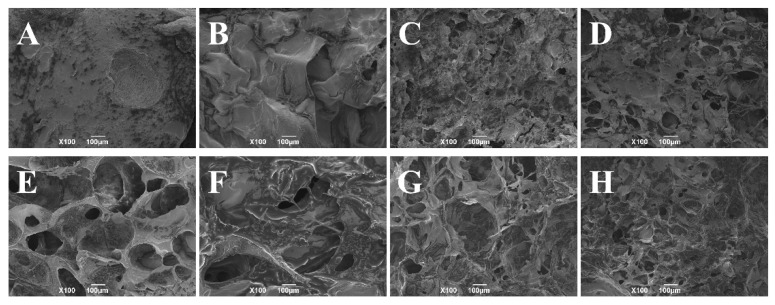
SEM images of konjac emulgel-based fat analogue (KGFA) with different ethanol content before and after freeze-thaw treatment. (**A**,**E**): 0% ethanol; (**B**,**F**): 2% ethanol; (**C**,**G**): 6% ethanol; (**D**,**H**): 8% ethanol.

**Table 1 foods-11-03173-t001:** Syneresis (%), heating loss (%), soybean oil content (wt%) and pH of pork backfat (PBF) and konjac emulgel-based fat analogue (KGFA) with different ethanol content after freeze-thaw treatment.

Samples	Syneresis (%)	Heating Loss (%)	pH	Soybean Oil Content(wt%)
Pork backfat	——	1.01 ± 0.15 ^c^	6.30 ± 0.08 ^d^	——
0	14.18 ± 0.77 ^a^	9.78 ± 0.71 ^a^	8.35 ± 0.04 ^c^	8.00
2	12.58 ± 0.47 ^b^	5.41 ± 0.58 ^b^	8.41 ± 0.05 ^c^	8.00
6	9.54 ± 0.72 ^c^	4.56 ± 0.22 ^b^	8.61 ± 0.04 ^b^	8.00
8	10.63 ± 0.46 ^c^	4.53 ± 0.62 ^b^	8.76 ± 0.06 ^a^	8.00

Mean ± SD. Different letters indicate significant differences in the same column (*p* < 0.05).

**Table 2 foods-11-03173-t002:** Colour parameters of pork backfat (PBF) and konjac emulgel-based fat analogue (KGFA) with different ethanol content before and after freeze-thaw treatment.

Samples	L*	a*	b*	Chroma	∆E	Whiteness Index	Browning Index
Pork backfat	58.73 ± 0.76 ^a^	−1.88 ± 0.10 ^c^	7.56 ± 1.01 ^a^	7.79 ± 1.22 ^a^	—	57.99 ± 1.04 ^a^	10.95 ± 2.30 ^a^
KGFA before freezing							
0	44.38 ± 0.81 ^c^	1.13 ± 0.42 ^a^	1.21 ± 0.16 ^b^	1.68 ± 0.16 ^b^	15.98 ± 0.86 ^a^	44.36 ± 0.81 ^c^	4.48 ± 0.39 ^b^
2	47.97 ± 0.06 ^b^	−0.16 ± 0.01 ^b^	−2.67 ± 0.16 ^c^	2.67 ± 0.16 ^b^	14.95 ± 0.15 ^a^	47.91 ± 0.06 ^b^	5.43 ± 0.28 ^c^
6	49.09 ± 1.58 ^b^	−0.36 ± 0.05 ^b^	−2.28 ± 0.11 ^c^	2.30 ± 0.11 ^b^	13.87 ± 1.18 ^ab^	49.05 ± 1.58 ^b^	4.85 ± 0.30 ^c^
8	49.83 ± 0.95 ^b^	−0.30 ± 0.03 ^b^	−1.26 ± 0.04 ^c^	1.30 ± 0.02 ^b^	12.64 ± 0.69 ^b^	49.81 ± 0.95 ^b^	2.82 ± 0.07 ^c^
Pork backfat	58.73 ± 0.76 ^a^	−1.88 ± 0.10 ^c^	7.56 ± 1.01 ^a^	7.79 ± 1.22 ^a^	—	57.99 ± 1.04 ^a^	10.95 ± 2.30 ^a^
KGFA after freezing							
0	32.72 ± 0.11 ^d^	0.80 ± 0.05 ^a^	−0.12 ± 0.06 ^b^	0.80 ± 0.06 ^c^	27.25 ± 0.13 ^a^	32.72 ± 0.11 ^d^	1.37 ± 0.06 ^b^
2	43.10 ± 0.93 ^c^	−0.36 ± 0.07 ^b^	−3.81 ± 0.15 ^c^	3.82 ± 0.16 ^b^	18.52 ± 2.06 ^b^	42.97 ± 0.93 ^c^	8.69 ± 0.59 ^a^
6	54.23 ± 0.98 ^b^	−0.40 ± 0.13 ^b^	−1.11 ± 0.16 ^b^	1.18 ± 0.18 ^c^	9.91 ± 0.34 ^c^	54.22 ± 0.97 ^b^	2.46 ± 0.38 ^b^
8	53.79 ± 1.23 ^b^	−0.17 ± 0.08 ^b^	−0.86 ± 0.24 ^b^	0.88 ± 0.25 ^c^	9.94 ± 0.80 ^c^	53.79 ± 1.24 ^b^	1.75 ± 0.57 ^b^

Mean ± standard deviation. For samples with different processing methods (before and after freeze-thaw treatment), different letters indicate significant differences in the same column (*p* < 0.05).

## Data Availability

The data presented in this study are available on request from the corresponding author.

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
