# Peer review of "Effect of Ethanol on Preparation of Konjac Emulgel-Based Fat Analogue by Freeze-Thaw Treatment"

_foods, 2022, doi:10.3390/foods11203173_

Round 1

Reviewer 1 Report

Hi dear

This article "Effect of Ethanol on Preparation of Konjac Gel Based Fat Analogue by Freeze-thaw Treatment” was revised and has a novelty and I recommend it for publication after consideration of the following comments.

Title: It is perfect.

Abstract:

·       The type of statistical design used in this research should be mentioned.

·       Treatments are not well defined.

·       Abstract is not comprehensive and complete please explain it more in terms of results provided.

Keywords: Please choose keywords other than the main words of the title. In this case, other researchers can find your article by searching a wide range of words through databases. I propose another keywords as the follow:

konjac gel; ethanol; pork backfat; Fat analogue; mechanical Textural properties, Physicochemical properties

Introduction:

·        In the introduction, state the reason why substitute Pork backfat is used as well as the roll of thawing-freezing for gel providing

·        The treatments used in the research should be stated in the last paragraph of the introduction.

Materials: It is complete and appropriate.

Methodology:

·       Nearly all the methods do not have references. Please cite https://doi.org/10.1002/fsn3.2050“for Line 83-85 (2.6. Colour measurement). Please express and provide chroma, Hue angle, ΔE, Browning index etc. as the references pointed to it.

·       Please cite https://doi.org/10.1111/jfpp.15311 “for Line 86-94 (2.7. Mechanical properties).

·       Statistical analysis don’t exist why?

·       Nearly all the methods do not have references.

·       “Water and fat binding properties” very bad expression please consider with more detail.

·       Line 67-82: The way of expressing the method of measuring the parameters has a scientific flaw. Please take help from the following article for the correct way of expressing it, so that the standard number of the working method should be clearly stated (https://doi.org/10.1590/fst.60820) or (https://doi.org/10.1590/fst.52120).

 Results:

·       All Tables and Figs: The alphabetical statistical letters for the means should all be modified such that the greatest number has the letter a and as the numbers go lower, letters b, c etc.

·       What is the necessity of conducting research tests before and after freezing? You used freezing only once and that was to make a better gel, please explain and even mention it in the introduction

·       The colors before and after freezing should be synchronized in all shapes

·       In Fig 5: P22 etc.? all Tables and Figs please provide self-explanatory

Discussion:

·       Discussion text must grammar improve and in some cases it is very weak and maybe there is no discussion at all.

·       Discussions are very, very poorly expressed and sometimes non-existent, for example:

3.5. Morphological analysis & 3.4. LF-NMR analysis etc.

Conclusions:

·       Conclusion is very general, try to make it more scientific, comprehensive and concise in detail, especially.

·       The conclusion is very similar to the set abstract. Please rewrite it and write a more favorable conclusion with scientific details and numbers and statistics.

References: It is OK.

The article has many flaws in express and concept of English, it is suggested to be revised in a scientific and native way.

Author Response

Reviewer 1

  1. This article "Effect of Ethanol on Preparation of Konjac Gel Based Fat Analogue by Freeze-thaw Treatment”was revised and has a novelty and I recommend it for publication after consideration of the following comments.

Title: It is perfect.

Response: Thank you very much for your valuable suggestions. We have revised the manuscript based on the comments.

  1. Abstract: The type of statistical design used in this research should be mentioned.

Response: Thanks for your kind advice. We have added the type of statistical design in the abstract of the revised manuscript.

  1. Abstract: Treatments are not well defined.

Response: Thanks for your valuable suggestions. We have added the definition of the treatment in the abstract of the revised manuscript.

  1. Abstract: Abstract is not comprehensive and complete please explain it more in terms of results provided.

Response:We deeply appreciate your valuable comments.We have revised the abstract to provide a more comprehensive interpretation of the experimental results.

  1. Keywords: Please choose keywords other than the main words of the title. In this case, other researchers can find your article by searching a wide range of words through databases. I propose another keywords as the follow: konjac gel; ethanol; pork backfat; Fat analogue; mechanical Textural properties, Physicochemical properties.

Response: Thanks for you suggestions. We have modified the keywords according to your advice.

  1. Introduction: In the introduction, state the reason why substitute Pork backfat is used as well as the roll of thawing-freezing for gel providing.

Response:Thank you very much for your suggestion, in fact we have explained the reasons for using freeze-thawed konjac gel instead of pork backfat in the introduction.The reasons are as follows:

Konjac gel was an ideal material for simulating pork backfat, but its texture was soft and had a certain gap with pork backfat. Freeze-thaw treatment was an effective method to improve the texture of konjac gel, but it will seriously damage the network structure of the gel, resulting in severe water separation and abnormal increase in gel hardness. So in this study, we tried to combine a poor solvent (ethanol) with a good solvent (water) to affect the assembly during the freezing process by influencing the state of the glucomannan molecular chain.

  1. Introduction: The treatments used in the research should be stated in the last paragraph of the introduction.

Response: Thanks for your kind advice. We have supplemented the explanation of the treatments in the last paragraph of the introduction.

  1. Materials: It is complete and appropriate.

Methodology: Nearly all the methods do not have references. Please cite https://doi.org/10.1002/fsn3.2050“for Line 83-85 (2.6. Colour measurement). Please express and provide chroma, Hue angle, ΔE, Browning index etc. as the references pointed to it.

Response: Thank you very much for your valuable suggestion. We have cited https://doi.org/10.1002/fsn3.2050 for 2.6. Colour measurement, and we also have provided chroma,ΔE, Browning index and whiteness index in Table 2.

  1. Methodology: Please cite https://doi.org/10.1111/jfpp.15311 “for Line 86-94 (2.7. Mechanical properties).

Response: Thanks for your suggestion. We have cited the reference https://doi.org/10.1111/jfpp.15311 in section 2.7 Mechanical properties.

  1. Methodology: Statistical analysis don’t exist why?

Response: Thank you for finding this error, we have added the section 2.10 statistical analysis. 

  1. Methodology: Nearly all the methods do not have references.

Response: Thanks for your kind advice. We have cited some references based on your suggestion.

  1. Methodology: “Water and fat binding properties” very bad expression please consider with more detail.

Response: Thanks for your question and suggestion. I'm so sorry for this expression. There is an expression of water and fat binding properties in this reference “doi:10.1016/j.foodhyd.2011.04.007”, and we refer to this article so we express it like this. Do you think it needs to be changed, if so, I will continue to modify it later. 

  1. Methodology: Line 67-82: The way of expressing the method of measuring the parameters has a scientific flaw. Please take help from the following article for the correct way of expressing it, so that the standard number of the working method should be clearly stated (https://doi.org/10.1590/fst.60820) or (https://doi.org/10.1590/fst.52120).

Response: Thanks for your valuable suggestion. We have revised the expression of the method of measuring the parameters.

2.3. Water and fat binding properties

The initial weight of the sample was weighed and recorded as m0, and then the sample was placed in a tube and heated at 70 °C for 30 min in a water bath. After heating, take out the emulgel and pour out the water and fat, then weigh the emulgel again and record  as m. Heating loss (water and fat loss) was calculated by the following equuation:

Heating loss(%)=(m-m0)/m0*100

2.4. Syneresis rate measurement

The weight of the samples before freezing was recorded as A (g), and the weight of the samples after freezing was recorded as B (g). The water separation rate can be calculated by the following formula.Syneresis rate was calculated by the following equation:

Syneresis(%) = (A-B) / A × 100

2.5. Measurement of pH

The samples were mixed with distilled water at a ratio of 1:10 (g/mL), then filtered with a funnel. Finally, the pH value was measured at room temperature with a Radiome-ter model PHM 93 pH meter (Meterlab, Copenhagen, Denmark).

  1. Results:All Tables and Figs: The alphabetical statistical letters for the means should all be modified such that the greatest number has the letter a and as the numbers go lower, letters b, c etc.

Response: Thanks for your kind advice.We have revised the alphabetical statistical letters for all means so that the largest number has the letter a, as the numbers go smaller, the letters b, c, etc.

  1. Results:What is the necessity of conducting research tests before and after freezing? You used freezing only once and that was to make a better gel, please explain and even mention it in the introduction.

Response:Thanks for your question and suggestion. The research tests before and after freezing are mainly to illustrate the effect of freeze-thaw treatment by comparison and to prove that freeze-thaw treatment can improve the texture properties of konjac gel. Therefore, it is indicted that freeze-thaw treatment is one of the necessary means to prepare konjac gel based fat analogue.

The freeze-thaw treatment for konjac gel was only one cycle in this study, because multiple cycles would lead to severe water separation and damage to the network structure. I have added this in the introduction.

  1. Results:The colors before and after freezing should be synchronized in all shapes.

Response:Thanks for your suggestion. We have revised the Table 2 to make them synchronized.

  1. Results:In Fig 5: P22 etc.? all Tables and Figs please provide self-explanatory.

Response: Thank you for finding this error. We have supplemented the explanation in Figure 5.  P2b, P21, P22 and P23 were the corresponding area fractions of T2b, T21, T22 and T23 in Figure 4.

  1. Discussion:Discussion text must grammar improve and in some cases it is very weak and maybe there is no discussion at all. Discussions are very, very poorly expressed and sometimes non-existent, for example: 5. Morphological analysis & 3.4. LF-NMR analysis etc.

Response: Thanks for your suggestion. I have revised the discussion section and rewrited 3.4. LF-NMR analysis and 3.5. Morphological analysis.

3.4. LF-NMR analysis: LF-NMR is a testing technique that can be used to measurement the degree of freedom and distribution of water in gel systems. The shorter the relaxation time, the tighter the binding of water molecules and the weaker the mobility of water molecules. T2b, T21, T22, and T23 were the relaxation times of different states of water in the gel. T2b represents water closely related to macromolecules with a relaxation time of 1-10ms, T21 represents water that does not flow easily, with a relaxation time of 10-100ms, T22 and T23 represent free water, with a relaxation time of 100-1000ms and 1000- 10000ms.

As shown in figure 4, a stronger signal was observed for the T22 component when the ethanol content was 0 wt%, which indicated that the free water ratio of the konjac emulgel based fat analogue was higher at this time. However the signal intensity of the T22 component was significantly reduced after adding 2 wt% ethanol, indicating that the different state water has changed. In addition, it can be seen that the signal intensity of T22 component decreases continuously, and T21 and T22 move to a lower relaxation time with the increase of ethanol content, indicating that the fluidity of water in the konjac emulgel based fat analogue decreased and free water gradually turned into bound water.

3.5. Morphological analysis: The proportion of the T2 distribution of konjac emulgel are shown in Figure 5. P21 in the unfrozen emulgels increased gradually with the increase of ethanol content, while P22 decreased, indicating that ethanol can change the water distribution, and free water was continuously converted into immobilized water. When the ethanol content were 0 wt% and 2 wt%,  P23 increased significantly, and P21 decreased after freezing, indicating that the degree of freedom of water increased and water migrated to free water. However, when the ethanol content were 6 wt% and 8 wt%, P23 did not increase significantly after freezing, and the change in water distribution was small, indicating that konjac emulgel based fat analogue prepared with this ethanol content had better stability.

  1. Conclusions:Conclusion is very general, try to make it more scientific, comprehensive and concise in detail, especially. The conclusion is very similar to the set abstract. Please rewrite it and write a more favorable conclusion with scientific details and numbers and statistics.

Response:Thank you very much for your valuable suggestion. We have rewrited the conclusions based on your comments.

The current study results showed that it is feasible to use ethanol to modulate the textural properties of konjac emulgel during freeze-thaw processes for the preparation of konjac emulgel based fat analogue. The method solved the problem of the gap between the texture of traditional konjac emulgel and pork backfat. Adding 6% ethanol and combining with one freeze-thaw treatment cycle can obtain a emulgel with similar texture properties to pork backfat, and its mechanical textural properties such as hardness, chewiness, and gel strength were similar to those of pork backfat. At the same time, the konjac emulgel based fat analogue also had good thermal stability with a small heat lossing of 4.56%. Whether this scheme had the properties of generalization remained to be revealed and the specific mechanism of changes in emulgelation and freezing remained to be revealed in the next step.

  1. References: It is OK.

The article has many flaws in express and concept of English, it is suggested to be revised in a scientific and native way.

Respnse: Thanks for your valuable suggestion. We have revised the express and concept of English, hopefully the previous flaws have been corrected.

Reviewer 2 Report

The paper “Effect of Ethanol on Preparation of Konjac Gel Based Fat Analogue by Freeze-thaw Treatment” describes the potential use of a konjac gel as a fat replacer. Gel properties are modified using ethanol to mimic better the characteristics of a pork fat; results seem promising, even if further investigation is necessary to understand the phenomena involved in konjac gelation and to explore potential practical uses.

I have some concerns about the use of the term “gel” because in materials and method section the authors report that soybean oil is present in formulation, even if the adopted amount is not reported. This means that they are preparing emulsions or, better, emulgels because the aqueous phase is gelled. As a consequence, probably, the title should be modified (“gel” could be replaced by “emulgel”) and also the Introduction and the Discussion should be modified to take into account that the authors are investigating emulgels (the term “gel” should be replaced by “emulgel” throughout the manuscript).

 Here you find some further specific comments:

Introduction: this section is very concise, I suggest to enlarge it describing, briefly, the results of previous studies where konjac gels (or emulgels) were used to replace fats. This will help in understanding the previous problems and the innovation introduced by the present work.

Line 57: Why Na2CO3 is used in solution?

Line 58: sample composition should be clearly reported, I was not able to understand the oil amount in samples; it could be added to Table 1.

Line 59: details about magnetic stirrer (model and manufacturer) should be added.

Line 60: So far, the authors discussed about gels, anyway here we discover that soybean oil is present and therefore emulgels were prepared, see my previous comments.

Line 60: Why was ethanol mixed with oil instead of adding it directly to the water phase?

Line 61: maybe “sol” could be replaced by “emulsion”

Line 63: the thermal history (i.e. heating and cooling rate) could be relevant and in the present case it was not controlled. It could be useful to give the mold dimension to help the reader in understanding how heating and cooling were fast (or slow).

Line 66: did the authors check that, after 4 hours, temperature at sample core was 25°C?

Line 81-82: was this procedure suitable for pH determination of pork backfat? I suppose that fat did not dissolve in water.

Line 90: sample dimensions should be given

Line 95: the term “HDP-BSW device” is not enough for a reader not expert in SMS devices. Few additional details could be useful to understand that it is a blade.

Line 104-106: further details about the adopted procedure are necessary. Did they use a gold coating? Did they use low vacuum conditions? Did they use a cryo-SEM? Did they prepare the sample before the test (for instance a fast freezing in liquid nitrogen)?

Line 147-149: this seems a speculation. Did the authors have any evidence about the potential increase in polymer interactions caused by crystallites?

Line 183-187: further explanation about the different relaxation times (i.e. T21, T22, T2b, T23), their meaning and how they were obtained from experimental T2 values has to be given.

Author Response

Responses to Reviewer 2

  1. The paper “Effect of Ethanol on Preparation of Konjac Gel Based Fat Analogue by Freeze-thaw Treatment” describes the potential use of a konjac gel as a fat replacer. Gel properties are modified using ethanol to mimic better the characteristics of a pork fat; results seem promising, even if further investigation is necessary to understand the phenomena involved in konjac gelation and to explore potential practical uses.

I have some concerns about the use of the term “gel” because in materials and method section the authors report that soybean oil is present in formulation, even if the adopted amount is not reported. This means that they are preparing emulsions or, better, emulgels because the aqueous phase is gelled. As a consequence, probably, the title should be modified (“gel” could be replaced by “emulgel”) and also the Introduction and the Discussion should be modified to take into account that the authors are investigating emulgels (the term “gel” should be replaced by “emulgel” throughout the manuscript).

Response: Thank you very much for your suggestion. In the process of preparing the sample, we added 8 wt% soybean oil, and it is really inappropriate to use the term “gel”. We have replaced the term “gel” throughout the manuscript with “emulgel” based on your comments.

  1. Introduction: this section is very concise, I suggest to enlarge it describing, briefly, the results of previous studies where konjac gels (or emulgels) were used to replace fats. This will help in understanding the previous problems and the innovation introduced by the present work.

Response:Thanks for your valuable suggestion.We have revised the introduction based on your comment to add the results of previous studies on fat replacement with konjac gels (or emulgels).

  1. Line 57: Why Na2CO3is used in solution?

Response:Thank you for your question. Because the preparation of konjac glucomannan into a thermally irreversible gel requires an alkaline environment, the characteristic property of thermal irreversibility of the gel is the key to prepare the konjac emulgel based fat analogue, which can ensure that the emulgel does not melt under high temperature conditions. Sodium carbonate is added to make the solution alkaline.

  1. Line 58: sample composition should be clearly reported, I was not able to understand the oil amount in samples; it could be added to Table 1.

Response: Thank you very much for your advice, and I am very sorry for the unclear description of the sample composition. The soybean oil content in the sample is 8 wt%, we have added the soybean oil content in Table 1.

  1. Line 59: details about magnetic stirrer (model and manufacturer) should be added.

Response: Thanks for your suggestion. We have added model and manufacturer of magnetic stirrer.

  1. Line 60: So far, the authors discussed about gels, anyway here we discover that soybean oil is present and therefore emulgels were prepared, see my previous comments.

Response: Thanks for your kind advice. We have replaced the term “gel” with “emulgel” based on your previous comments.

  1. Line 60: Why was ethanol mixed with oil instead of adding it directly to the water phase?

Response: Thank you for your question. The fact that oil was added directly to the water phase resulted in many large oil droplets on the surface of the emulgel, which indicates a lot of oil loss. After continuous experiments, we found that this problem can be solved by mixing ethanol and oil.

  1. Line 61: maybe “sol” could be replaced by “emulsion”

Response: Thanks for your suggestion.We have replaced “sol” with “emulsion”.

  1. Line 63: the thermal history (i.e. heating and cooling rate) could be relevant and in the present case it was not controlled. It could be useful to give the mold dimension to help the reader in understanding how heating and cooling were fast (or slow).

Response: Thank you very much for your valuable suggestion. The mold dimension is 170mm*95mm*20mm. We have added the mold dimension in the revised manuscript.

  1. Line 66: did the authors check that, after 4 hours, temperature at sample core was 25°C?

Response: Thank you for your question. Yes, I checked, temperature at sample core was 25°C after 4 hours.

  1. Line 81-82: was this procedure suitable for pH determination of pork backfat? I suppose that fat did not dissolve in water.

Response: Thanks for your question. We refer to the method for the determination of pH of pork backfat in the article “doi:10.1016/j.foodhyd.2011.04.007”. The specific measurement is as follows: The pH was determined on a Radiometer model PHM 93 pHmeter (Meterlab, Copenhagen, Denmark) at room temperature on homogenates of samples in distilled water in a ratio 1:10 (w/v). Three replicates were performed for each formulation.

  1. Line 90: sample dimensions should be given

Response: Thanks for your kind advice.The sample dimension is 30mm*30mm*20mm, and we have added this information in the revised manuscript.

  1. Line 95: the term “HDP-BSW device” is not enough for a reader not expert in SMS devices. Few additional details could be useful to understand that it is a blade.

Response: Thanks for your suggestion. I have added detailed instructions about the HDP-BSW.

  1. Line 104-106: further details about the adopted procedure are necessary. Did they use a gold coating? Did they use low vacuum conditions? Did they use a cryo-SEM? Did they prepare the sample before the test (for instance a fast freezing in liquid nitrogen)?

Response: Thank you very much for your valuable suggestion. I have added the further details about the adopted procedure. The samples were put into liquid nitrogen for instance a fast freezing and then placed in a vacuum freeze dryer to obtain lyophilized samples. The samples were sprayed with gold using an ion sputtering device under low vacuum conditions. Finally, the surface morphology of the samples was observed by scanning electron microscopy(JSM-6390LV, Jeol, Japan) at an accelerating voltage of 5 Kv and a magnification of 100×.

  1. Line 147-149: this seems a speculation. Did the authors have any evidence about the potential increase in polymer interactions caused by crystallites?

Response: Thank you very much for your question, I have not studied the formation and changes of crystallites during freezing, so there is no evidence that it increases polymer interactions. This sentence mainly refers to some related research results that have been reported.

  1. Line 183-187: further explanation about the different relaxation times (i.e. T21, T22, T2b, T23), their meaning and how they were obtained from experimental T2 values has to be given.

Response: Thanks for your suggestion. We have added the further explanation about the different relaxation times.

T2b, T21, T22, and T23 were the relaxation times of different states of water in the gel. T2b represents water closely related to macromolecules with a relaxation time of 1-10ms, T21 represents water that does not flow easily, with a relaxation time of 10-100ms, T22 and T23 represent free water, with a relaxation time of 100-1000ms and 1000- 10000ms.

Round 2

Reviewer 1 Report

The manuscript was revised according to the suggestions and it is acceptable in this format.

Reviewer 2 Report

The paper was amended according to my comments. It is now suitable for publication